# Comparison of Diagnostic Performance between Classic and Modified Abbreviated Breast MRI and the MRI Features Affecting Their Diagnostic Performance

**DOI:** 10.3390/diagnostics14030282

**Published:** 2024-01-27

**Authors:** Subin Lee, Eun Jung Choi, Hyemi Choi, Jung Hee Byon

**Affiliations:** 1Department of Radiology, Research Institute of Clinical Medicine of Jeonbuk National University-Biomedical Research Institute of Jeonbuk National University Hospital, Jeonbuk National University Medical School, Jeonju 54907, Jellabuk-Do, Republic of Korea; subin2787@gmail.com (S.L.); cejcej80@jbnu.ac.kr (E.J.C.); 2Department of Statistics and Institute of Applied Statistics, Jeonbuk National University, Jeonju 54896, Jellabuk-Do, Republic of Korea; hchoi@jbnu.ac.kr; 3Department of Radiology, Ulsan University Hospital, University of Ulsan College of Medicine, Ulsan 44610, Republic of Korea

**Keywords:** breast neoplasm, magnetic resonance imaging, screening, area under the curve, sensitivity, specificity

## Abstract

Abbreviated breast magnetic resonance imaging (AB-MRI) has emerged as a supplementary screening tool, though protocols have not been standardized. The purpose of this study was to compare the diagnostic performance of modified and classic AB-MRI and determine MRI features affecting their diagnostic performance. Classic AB-MRI included one pre- and two post-contrast T1-weighted imaging (T1WI) scans, while modified AB-MRI included a delayed post-contrast axial T1WI scan and an axial T2-weighted interpolated scan obtained between the second and third post-contrast T1WI scans. Four radiologists (two specialists and two non-specialists) independently categorized the lesions. The MRI features investigated were lesion size, lesion type, and background parenchymal enhancement (BPE). The Wilcoxon rank-sum test, Fisher’s exact test, and bootstrap-based test were used for statistical analysis. The average area under the curve (AUC) for modified AB-MRI was significantly greater than that for classic AB-MRI (0.76 vs. 0.70, *p* = 0.010) in all reader evaluations, with a similar trend in specialist evaluations (0.83 vs. 0.76, *p* = 0.004). Modified AB-MRI demonstrated increased AUCs and better diagnostic performance than classic AB-MRI, especially for lesion size > 10 mm (*p* = 0.018) and mass lesion type (*p* = 0.014) in specialist evaluations and lesion size > 10 mm (*p* = 0.003) and mild (*p* = 0.026) or moderate BPE (*p* = 0.010) in non-specialist evaluations.

## 1. Introduction

Breast magnetic resonance imaging (MRI) shows high sensitivity and specificity for breast cancer screening and outperforms mammography, digital breast tomosynthesis, and ultrasonography (US) [1]. The American Cancer Society guidelines highly recommend breast MRI screening as a supplementary screening tool for women with an approximately 20–25% greater lifetime risk of breast cancer [2]. The American College of Radiology (ACR) also recommends it for women with a history of breast cancer and dense tissue [3]. Nevertheless, the use of breast MRI as a supplementary screening tool is limited by several disadvantages, such as the high cost and long acquisition and interpretation times [2,4]. To overcome these limitations, abbreviated breast MRI (AB-MRI) emerged in 2014 [5,6], showing diagnostic accuracy and cancer detection rates comparable to those of conventional full-protocol MRI [5,7].

However, AB-MRI also has several inherent challenges, such as high recall and false-positive rates, owing to the lack of delayed contrast-enhanced images [5,7]. Furthermore, since Kuhl et al. first introduced AB-MRI [5], AB-MRI protocols have not been standardized. The most commonly described AB-MRI, hereafter referred to as the classic AB-MRI, consists of the most essential sequences, including pre-contrast T1-weighted imaging (T1WI) and a single post-contrast-enhanced T1WI scan. However, this protocol does not fulfill the ACR breast MRI accreditation requirements and may not yield the high sensitivity and specificity observed in full-protocol MRI since it does not include kinetic-curve analyses using delayed contrast-enhanced images and fat-suppressed T2-weighted imaging (T2WI) [3,6]. Although several studies have evaluated various AB-MRI as supplementary screening tools, an AB-MRI that provides kinetic information while ensuring a shorter image acquisition time has not been described to date [8,9].

To fulfill the ACR breast MRI accreditation requirements and overcome the abovementioned limitations of the classic AB-MRI protocol, we implemented a modified AB-MRI protocol that added one T2WI and one delayed-phase post-contrast T1WI scan to the classic AB-MRI protocol, maintaining an acceptable acquisition time of 10 min since the T2WI scan was interposed between the early phase and the delayed phase of the post-contrast T1WI scan. Although this modified AB-MRI can fulfill the ACR breast MRI accreditation requirements and provide more information than the classic AB-MRI protocol, the effects of adding one T2WI and one delayed-phase post-contrast T1WI scan have not been investigated until now.

Thus, the purpose of this study was to compare the diagnostic performance of the modified AB-MRI protocol with that of the classic AB-MRI and to determine the MRI features affecting the diagnostic performance of both.

## 2. Materials and Methods

### 2.1. Study Population

This retrospective study was approved by the institutional review board, and the requirement for informed consent was waived. Between August 2019 and January 2022, a retrospective search of our institution’s database identified 140 lesions that were evaluated with the modified AB-MRI in women with a personal history of breast cancer surgery. From this population, 122 lesions evaluated in 118 MRI examinations in 105 women were included in this study. The inclusion criteria were as follows: (a) lesions classified as Breast Imaging Reporting and Data System (BI-RADS) final assessment category 4 or 5 and (b) lesions that were recalled for second-look ultrasound after assessment with the modified AB-MRI. The exclusion criteria were as follows: (a) lesions that were not biopsied after the second-look ultrasound and did not have at least one year of follow-up data (*n* = 10); (b) lesions that were not evaluated by the modified AB-MRI protocol (*n* = 5); and (c) lesions that showed low AB-MRI image quality due to artifacts or incomplete fat suppression (*n* = 3) (Figure 1).

Lesions were histologically confirmed using excisional biopsy with US (*n* = 49) or mammography-guided (*n* = 3) wire localization or without localization (*n* = 3) or surgical excision after percutaneous US-guided (*n* = 15) biopsy.

### 2.2. Breast MRI Examination Technique

All breast MRI examinations were performed using one of two 3.0-T systems (Magnetom Skyra or Magnetom Verio; Siemens Healthineers, Erlangen, Germany) with a dedicated breast coil and prone patient positioning. The modified AB-MRI included one pre-contrast and three post-contrast axial T1-weighted fat-suppressed spoiled gradient-echo scans (repetition time/echo time (TR/TE), 4.3–4.4/1.6–1.7 ms; matrix size, 448 × 354; field of view (FOV), 340 × 340 mm; flip angle, 10°; section thickness/gap, 1.0 mm/0 mm). An axial T2-weighted fat-suppressed fast spin-echo scan was interpolated between the second and third post-contrast axial T1WI scans (Magnetom Verio: TR/TE, 5300-5660/61; matrix size, 512 × 358; FOV, 340 × 340 mm; section thickness/gap, 2.0 mm/0–0.3 mm, Magnetom Skyra: TR/TE, 5800/63; matrix size, 448 × 269; FOV, 340 × 340 mm; section thickness/gap, 2.0 mm/0 mm). Postprocessing procedures, including subtraction, sagittal multiplanar reconstruction (MPR), and sagittal, coronal, or axial maximum intensity projection (MIP), were performed. For dynamic contrast-enhanced MRI, one pre-contrast and three post-contrast dynamic images were acquired 50, 105, and 420 s after the administration of gadobutrol (Gadovist; Schering AG, Berlin, Germany) at a dose of 0.1 mmol/kg at a rate of 2 mL/s, followed by a 20 mL saline flush. The total acquisition time for the modified AB-MRI was 8–10 min.

The classic AB-MRI included one pre-contrast and two post-contrast axial T1-weighted fat-suppressed spoiled gradient-echo scans (TR/TE, 4.3–4.4/1.6–1.7; matrix size, 448 × 381; FOV, 340 × 340 mm; flip angle, 10°; section thickness/gap, 1.0 mm/0 mm). Two post-contrast T1WI scans were required to obtain the peak phase of post-contrast T1WI, without a pause after contrast injection. Postprocessing imaging procedures, including subtraction and sagittal, coronal, or axial MIP, were performed. The total acquisition time for classic AB-MRI was 2–3 min. The sequences required for each session (classic and modified AB-MRI) were selected from the modified AB-MRI and saved as anonymized Digital Imaging and Communication in Medicine (DICOM) files. A schematic comparison of the modified and classic AB-MRI used in this study is presented in Figure 2.

### 2.3. Breast MRI Analysis

Four radiologists participated as readers and completed a quality workshop before participation. They received a brief lecture and passed a certification test for the interpretation of both classic and modified AB-MRI images during AB-MRI reader training. Subsequently, the radiologists individually reviewed the images on an INFINITT PACS monitor (INFINITT Healthcare, Seoul, South Korea). All radiologists involved in this study were fully aware of the ACR BI-RADS lexicon for breast MRI [10]. We independently categorized the breast lesions as category 2 (benign), 3 (probably benign), 4 (suspicious; 4a, low suspicion; 4b, intermediate suspicion; 4c, moderate suspicion), or 5 (highly suggestive of malignancy) based on the fifth edition of the BI-RADS.

The readers, who had 1, 2, 11, and 16 years of experience in interpreting breast MRI, were blinded to the patients’ clinical information, images from other modalities, and histopathological results of the breast lesions using anonymized DICOM files saved in the research folder on the PACS. Only a history of breast cancer surgery was provided to the readers.

The readers independently interpreted all modified AB-MRI scans in random order six weeks after completing all classic AB-MRI evaluations. The reference standard was the histopathological result or a negative image obtained at the 1-year negative follow-up examination. In the second session, the readers were allowed to see their own category assessments from the first session but remained blinded to the histopathologic results and the category assessments of the other readers.

We investigated the following MRI features affecting the diagnostic performance of AB-MRI: lesion size (>10 mm vs. ≤10 mm), which was measured at the longest diameter [11]; lesion type (mass, non-mass enhancement (NME), or focus), as defined by the ACR BI-RADS lexicon for breast MRI (10); and background parenchymal enhancement (BPE), which was categorized as minimal, mild, moderate, and marked [10]. The T2 signal intensity (SI) of each lesion was first categorized as high (high T2 SI) or not high (low, isointense, or mixed T2 SI) for statistical purposes [12]. The MRI features were evaluated using subtraction images processed during the second dynamic phase, which was the peak phase. Kinetic-curve findings of the delayed phase were divided into three groups for analysis: persistent, plateau, or washout, as defined by the ACR BI-RADS [10]. The readers were categorized as specialists and non-specialists based on whether they were fellowship-trained breast radiologists [13].

### 2.4. Statistical Analysis

The readers’ agreement on BI-RADS final assessment based on both classic and modified AB-MRI was assessed using kappa (κ) statistics, and we used the following definitions: less than 0.20 indicates poor agreement; 0.21–0.40 indicates fair agreement; 0.41–0.60 indicates moderate agreement; 0.61–0.80 indicates good agreement; and 0.81–1.00 indicates very good agreement [14].

We compared the sensitivity, specificity, and area under the ROC curve (AUC) of the classic and modified AB-MRI for each reader according to the MRI features. Wilcoxon rank-sum test and Fisher’s exact test were performed for continuous and categorical variables, respectively. To assess the statistical differences among AUC values, we used bootstrap-based tests from the pROC R library for all pairwise comparisons [15]. Bootstrap percentile confidence intervals (CIs) for AUCs were computed on the basis of 2000 bootstrap samples. Statistical analyses were performed using SAS (version 9.1.3; SAS Institute, Cary, NC, USA). *p* values < 0.05 were considered to indicate statistical significance.

## 3. Results

This study included 122 lesions evaluated in 118 MRI examinations in 105 women (mean age, 51.3, 27–82 years). All 122 lesions were identified using breast MRI, and targeted US was performed for each lesion. The histopathologically confirmed benign and malignant lesions are detailed in Table 1. Of the 94 benign breast lesions, 52 lesions (55.3%) remained stable after a one-year follow-up and were categorized as benign.

The clinical and MRI findings of the benign and malignant lesions are shown in Table 2. The malignant lesions were larger than the benign lesions (median size, 9 mm vs. 14 mm; *p* = 0.008). Mass type was more frequent in the malignant lesions (57.1% vs. 31.9%; *p* = 0.005).

The overall interobserver agreement for the BI-RADS final assessment based on modified AB-MRI (k = 0.574, 95% confidence interval (CI): 0.551–0.586) was higher than those based on classic AB-MRI (k = 0.538, 95% CI: 0.524–0.551).

### 3.1. Performance of the Modified and Classic AB-MRIs

As detailed in Table 3, the modified AB-MRI showed a significantly higher average sensitivity (87.5% vs. 76.8%, *p* = 0.014) and average AUC (0.76 vs. 0.70, *p* = 0.010). When the readers were classified as non-specialists or specialists, the modified AB-MRI showed significantly higher sensitivity, specificity, and AUC in specialist evaluations (Table 3; Figure 3). In non-specialist evaluations, the modified AB-MRI showed higher sensitivity without significance, lower specificity, and higher AUC with borderline significance (91.1% vs. 76.8%, *p* = 0.156 for sensitivity; 35.1% vs. 48.4%, *p* = 0.018 for specificity; and 0.70 vs. 0.64, *p* = 0.05 for AUC; Table 3) (Figure 3).

### 3.2. MRI Features Affecting the Diagnostic Performance of Modified and Classic AB-MRI According to the Reader’s Experience

For average evaluation of all readers, lesion size > 10 mm was associated with a higher average AUC in the modified AB-MRI (Table 4). Mass lesion type was also associated with higher average sensitivity in the modified AB-MRI (93.8% vs. 82.8%, *p* = 0.022) (Table 5). However, no significant differences were observed regarding AUC in relation to NME (0.78 vs. 0.94, *p* = 0.182), focus (0.42 vs. 0.33, *p* = 0.071) (Table 5), and all levels of BPE (Table 6).

In non-specialist evaluations, lesion size > 10 mm was associated with a significantly higher average AUC for modified AB-MRI (0.68 vs. 0.56, *p* = 0.003), while lesion type was not associated with significant differences in the average AUC between the classic and modified AB-MRI (Table 5). However, in mild or moderate BPE, the average AUC for modified AB-MRI was higher than those for classic AB-MRI (0.74 vs. 0.58, *p* = 0.026 for mild BPE and 0.80 vs. 0.69, *p* = 0.010 for moderate BPE, respectively; Table 6).

For specialist evaluations, lesion size > 10 mm was associated with a higher AUC for modified AB-MRI (0.84 vs. 0.74, *p* = 0.018; Table 4), similar to mass lesion type (0.87 vs. 0.76, *p* = 0.014; Table 5). Moreover, mass lesion type was also associated with a higher sensitivity for modified AB-MRI (93.8% vs. 81.2%, *p* = 0.025; Table 5). However, the average AUC for the classic and modified AB-MRI did not differ in relation to the degree of BPE (Table 6).

## 4. Discussion

This study compared the diagnostic performance of two AB-MRI and identified the MRI features that affect the diagnostic performance. In this study, in comparison with the classic AB-MRI, the modified AB-MRI with additional T2WI and delayed-phase post-contrast T1WI scans tended to show higher average sensitivity (*p* = 0.014), comparable average specificity (*p* = 0.709), and a higher average AUC (*p* = 0.010) for all readers. Moreover, lesion size > 10 mm and the mass lesion type were related to the improved diagnostic performance of the modified AB-MRI in the specialist group, and lesion size > 10 mm and mild or moderate BPE were related to the improved performance in the non-specialist group.

In this study, the average sensitivity of the modified AB-MRI for all readers (87.5%) was greater than that of the classic AB-MRI (76.8%) and comparable to the previously reported value for the full MRI protocol (88.2%) [14]. According to the readers’ experience, the average sensitivity, specificity, and AUC of modified AB-MRI for specialists were significantly higher than those of classic AB-MRI, in contrast to the findings for non-specialists. Although studies comparing the diagnostic performance of AB-MRI and modified AB-MRI on the basis of reader experience are limited, several studies have compared the diagnostic performance of classic AB-MRI and full-protocol MRI in relation to the reader’s experience [16,17]. A previous study demonstrated that experienced readers showed significantly higher sensitivity and specificity than inexperienced readers [16]. In another study, less experienced readers reported more false-positive findings than experienced readers [17]. While more studies on modified AB-MRI focusing on improvements in diagnostic performance based on the reader’s experience are required to ensure the appropriate use of modified AB-MRI in clinical practice, the findings of the present study indicate that the inclusion of kinetic-curve analysis in classic AB-MRI could enhance the diagnostic performance, especially for experienced readers.

In our assessment of the MRI features affecting the diagnostic performance of modified and classic AB-MRI, lesion size ≤ 10 mm was not associated with significant differences between modified and classic AB-MRI in terms of sensitivity, specificity, and AUC, not only for all readers but also for the specialist group. In general, kinetic-curve features may not help determine whether small lesions (≤5 mm) are benign or malignant on MRI [13]. In another study, of the five false-negative lesions evaluated by AB-MRI based on single first post-contrast images, three were less than 1 cm in size, consistent with our results [15]. The sensitivity and AUC of the modified and classic AB-MRI also differed significantly in relation to the mass lesion type for all readers as well as a specialist group. Previous studies reported that AB-MRI incorporating both complete dynamic T1WI and T2WI scans showed non-inferior diagnostic performance to full-protocol MRI, especially for the mass lesion type rather than NME or focus (66.6%, 33.3%, and 0%, respectively), consistent with our results [13,15,18]. Another previous study reported that high BPE can reduce the accuracy of breast cancer detection and staging [19]. Consistent with our initial hypothesis, although specialists’ performance in both classic and modified AB-MRI assessments did not differ in relation to the level of BPE, non-specialists’ diagnostic performance with modified AB-MRI significantly increased with mild or moderate BPE. The masking effect of high BPE in detecting and differentiating benign and malignant breast lesions may have worsened the prognosis of breast cancer because of interval cancer. In this regard, modified AB-MRI can help non-specialists with less experience in interpreting breast MRI and yield a better diagnostic performance, especially with mild or moderate BPE.

This study had several limitations. First, it was a retrospective study conducted at a single institution and included lesions detected on breast MRI screening in women with a history of breast cancer surgery. Second, although BI-RADS recommends that BI-RADS category 3 lesions be downgraded to category 2 if no interval change is observed after a 2-year follow-up, we validated the diagnosis with follow-up imaging surveillance and medical records after 1 year based on a previous publication [20]. Third, modified AB-MRI in this study included T2WI interposed between the early and delayed-phase T1WI scans to simultaneously shorten the scanning time and gain the benefit of delayed-phase and T2WI scans. Although the influence of gadolinium contrast agents on T2WI in breast MRI remains unclear, the reasons can be inferred from a few previous reports [21,22,23]. Fourth, breast lesions assigned as BI-RADS category 4 or 5 on breast MRI or US were included in this study. Although these lesions might lead to overestimating sensitivity of breast MRI, the difference in diagnostic performance between classic and modified AB-MRI is likely to have been minimally impacted. Fifth, patients who received benign results based on histopathologic confirmation underwent relatively short-term follow-up imaging surveillance. To accurately evaluate the diagnostic performance of AB-MRI, further studies involving long-term follow-up imaging surveillance will be necessary. Finally, a cost-effectiveness analysis of the modified AB-MRI protocol was not performed, suggesting a need for further study to fully assess its effectiveness compared to the classic protocol.

In conclusion, modified AB-MRI demonstrated better diagnostic performance than classic AB-MRI, especially for lesions > 10 mm and mass-type lesions in the specialist group, and those >10 mm and showing mild or moderate BPE in the non-specialist group. Although further studies are needed, evaluation of the MRI features that affect the diagnostic performance of AB-MRI on the basis of the results of this study could enable the application of optimized and individualized AB-MRI, leading to improved diagnostic performance in the near future.

## Figures and Tables

**Figure 1 diagnostics-14-00282-f001:**
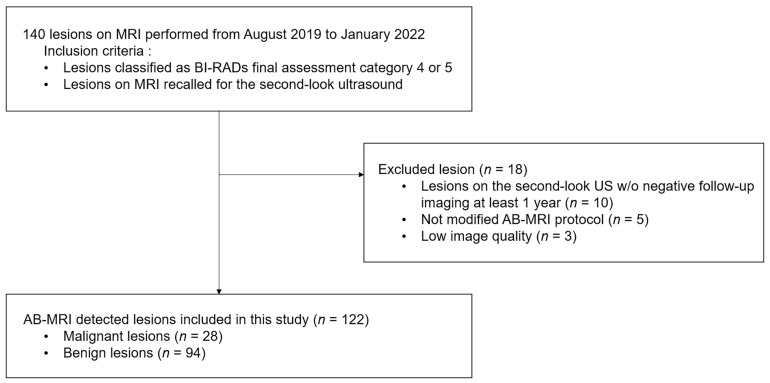
Flow chart of study sample selection. AB-MRI = abbreviated breast magnetic resonance imaging. BI-RADs = Breast Imaging Reporting and Data System. w/o = without.

**Figure 2 diagnostics-14-00282-f002:**
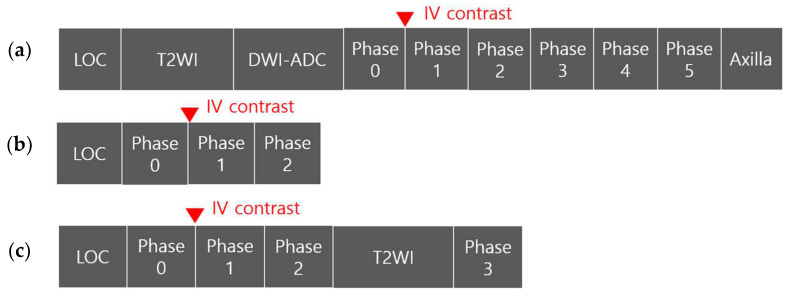
Schematic comparison of the modified and classic abbreviated breast magnetic resonance imaging (AB-MRI) used in this study. (**a**) The full protocol of breast MRI included the localizer, axial T2-weighted imaging (T1WI), diffusion-weighted image-apparent diffusion coefficient evaluation, a dynamic pre-contrast axial T1WI scan (phase 0) and five dynamic post-contrast axial T1WI scans (phase 1–5), and a sagittal delayed-enhanced axillary T1WI scan. (**b**) The classic AB-MRI protocol included the localizer and one pre-contrast (phase 0) and two post-contrast axial T1WI scans (phase 1–2). (**c**) The modified AB-MRI protocol consisted of the classic AB-MRI protocol with an interpolated T2-weighted image acquisition and a third post-contrast axial T1WI scan (phase 3).

**Figure 3 diagnostics-14-00282-f003:**
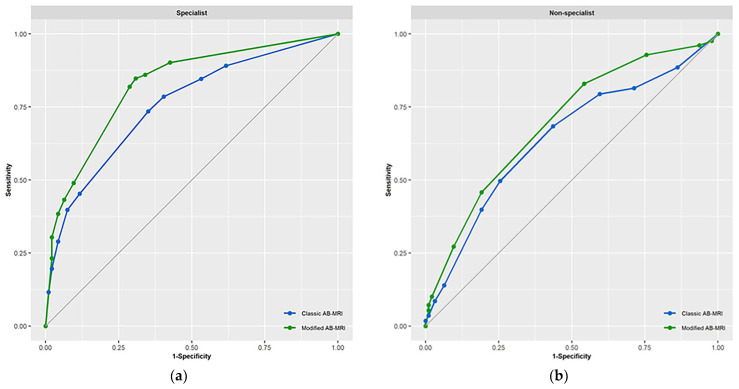
Comparison of diagnostic performance between classic and modified abbreviated breast MRI (AB-MRI) according to the reader’s experience. (**a**) The AUC of specialists was significantly higher for modified AB-MRI than for classic AB-MRI. (**b**) Non-specialists showed higher AUC values with borderline significance in comparisons between modified and classic AB-MRI protocols.

**Table 1 diagnostics-14-00282-t001:** Histopathologic features of 122 lesions.

Histopathologic Features	*n* (%)
Benign lesions	94 (100)
Fibrocystic change	16 (17.0)
Chronic inflammation	12 (12.8)
Fat necrosis	6 (6.4)
Xanthogranulomatous inflammation	3 (3.2)
Atypical ductal hyperplasia	1 (1.1)
Intraductal papilloma	1 (1.1)
Usual ductal hyperplasia	1 (1.1)
Atypical cells in lobules	1 (1.1)
Foreign body reaction	1 (1.1)
1-year negative follow-up	52 (55.3)
Malignant lesions	28 (100)
Invasive ductal cancer	20 (71.4)
Ductal carcinoma in situ	7 (25.0)
Metaplastic carcinoma	1 (3.6)

Data represent the number of lesions, and the values in parentheses are percentages.

**Table 2 diagnostics-14-00282-t002:** Comparison of AB-MRI characteristics between benign and malignant breast lesions.

AB-MRI Characteristics	Benign Lesions (*n* = 94)	Malignant Lesions (*n* = 28)	*p* Value
Median lesion size on MRI (mm) *	9 (3–68)	14 (3–49)	0.008
≤10 mm	58 (61.7)	10 (35.7)	0.018
>10 mm	36 (38.3)	18 (64.3)	
Lesion type	94 (100.0)	28 (100.0)	0.023
Focus	31 (33.0)	3 (10.7)	
Non-mass enhancement	33 (35.1)	9 (32.1)	
Mass	30 (31.9)	16 (57.1)	
Background parenchymal enhancement			0.968
Minimal	36 (38.3)	10 (35.7)	
Mild	30 (31.9)	10 (35.7)	
Moderate	25 (26.6)	7 (25.0)	
Marked	3 (3.2)	1 (3.6)	
Kinetic analysis of delayed phase			0.129
Persistent	26 (26.7)	5 (17.9)	
Plateau	21 (22.3)	3 (10.7)	
Washout	47 (50.0)	20 (71.4)	
T2 signal intensity **			0.307
Not-high group (hypo/iso/mixed)	70 (74.5)	24 (85.7)	
High group (hyper)	24 (25.5)	4 (14.3)	

Data represent the number of lesions, and the values in parentheses are percentages. * Numbers are median values, and the values in parentheses are ranges. MRI features were described and classified by two special radiologists in consensus who did not participate in the reader study of both AB-MRI protocols. AB-MRI, abbreviated breast magnetic resonance imaging. ** Definitions are as follows: low = low T2 SI, high = high, isointense, or mixed T2 SI compared to breast parenchyma.

**Table 3 diagnostics-14-00282-t003:** Performance of the four readers in assessing 122 breast lesions with the classic and modified abbreviated MRI protocols.

Reader	Sensitivity *			Specificity *			AUC **		
	Classic	Modified	*p*	Classic	Modified	*p*	Classic	Modified	*p*
Reader 1 (non-specialist)	82.1 (64.7–92.4)	92.9 (77.7–98.4)	0.424	40.4 (31.0–50.5)	24.5 (16.8–34.0)	<0.001	0.63 (0.51–0.74)	0.69 (0.58–0.79)	0.401
Reader 2 (non-specialist)	71.4 (53.1–84.9)	89.3 (73.1–96.6)	0.075	56.4 (46.3–66.0)	45.7 (36.0–55.8)	0.078	0.66 (0.53–0.78)	0.71 (0.60–0.80)	0.48
Reader 3 (specialist)	78.6 (60.7–90.0)	85.7 (68.8–94.6)	0.803	59.6 (49.5–69.0)	69.1 (59.3–77.6)	0.002	0.74 (0.63–0.84)	0.81 (0.72–0.90)	0.056
Reader 4 (specialist)	75 (56.9–87.5)	82.1 (64.7–92.4)	0.617	64.9 (54.9–73.8)	71.3 (61.5–79.5)	0.118	0.77 (0.67–0.86)	0.84 (0.75–0.91)	0.046
Average of non-specialists	76.8 (43.5–100.0)	91.1 (83.3–98.8)	0.156	48.4 (0–100.0)	35.1 (0–100.0)	0.018	0.64 (0.57–0.72)	0.70 (0.63–0.76)	0.050
Average of specialists	76.8 (64.9–88.7)	83.9 (72.5–95.4)	0.034	62.2 (51.2–73.2)	70.2 (61.9–78.5)	0.032	0.76 (0.66–0.85)	0.83 (0.74–0.91)	0.004
Average of all readers	76.8 (66.4–87.2)	87.5 (78.1–96.9)	0.014	55.3 (40.0–70.7)	52.7 (18.6–86.7)	0.709	0.70 (0.59–0.81)	0.76 (0.65–0.87)	0.010

* Numbers are percentages and 95% confidence intervals are in parentheses. ** Numbers in parentheses are the 95% confidence intervals of the AUC values. *p* values refer to the differences in the diagnostic performance between the classic and modified AB-MRI protocols. AUC = area under the receiver operating characteristic curve; *p* = *p* value; AB-MRI = abbreviated breast magnetic resonance imaging; classic = classic AB-MRI; modified = modified AB-MRI.

**Table 4 diagnostics-14-00282-t004:** Diagnostic performance of the four readers in assessing 122 breast lesions with the classic and modified abbreviated MRI protocols in relation to lesion size on MRI.

	Sensitivity *			Specificity *			AUC **		
	Classic	Modified	*p*	Classic	Modified	*p*	Classic	Modified	*p*
≤10 mm									
Reader 1 (non-specialist)	70.0 (40.1–89.6)	90.0 (60.4–99.0)	0.617	48.3 (35.9–60.8)	29.3 (19.1–41.9)	<0.001	0.64 (0.43–0.84)	0.67 (0.49–0.84)	0.798
Reader 2 (non-specialist)	80.0 (49.6–94.9)	80.0 (49.6–94.9)	0.617	58.6 (45.8–70.4)	44.8 (32.7–57.5)	0.080	0.70 (0.48–0.88)	0.63 (0.44–0.79)	0.464
Reader 3 (specialist)	70.0 (40.1–89.6)	70.0 (40.1–89.6)	0.617	74.1 (61.7–83.8)	79.3 (67.4–87.9)	0.790	0.74 (0.56–0.89)	0.74 (0.57–0.89)	0.978
Reader 4 (specialist)	40.0 (16.6–68.5)	50.0 (23.7–76.3)	0.998	81.0 (69.3–89.2)	84.5 (73.2–91.8)	0.901	0.72 (0.57–0.86)	0.77 (0.60–0.91)	0.369
Average of non-specialists	75.0 (55.7–94.3)	85 (62.8–100.0)	0.500	53.4 (26.7–80.2)	37.1 (0–81.4)	0.005	0.67 (0.54–0.80)	0.65 (0.52–0.78)	0.776
Average of specialists	55.0 (0–100)	60.0 (21.5–98.5)	0.500	77.6 (64.2–91.0)	81.9 (71.5–92.3)	0.275	0.73 (0.58–0.87)	0.76 (0.60–0.91)	0.477
Average of all readers	65.0 (37.3–92.7)	72.5 (45.3–99.7)	0.256	65.5 (43.6–87.4)	59.5 (18.1–100.0)	0.407	0.70 (0.56–0.83)	0.70 (0.56–0.84)	0.903
>10 mm									
Reader 1 (non-specialist)	88.9 (67.7–97.4)	94.4 (74.8–99.6)	0.998	27.8 (15.7–43.8)	16.7 (7.6–31.7)	0.261	0.520 (0.37–0.68)	0.65 (0.50–0.78)	0.142
Reader 2 (non-specialist)	66.7 (44.0–83.9)	94.4 (74.8–99.6)	0.022	52.8 (37.0–68.0)	47.2 (32.0–63.0)	0.956	0.60 (0.44–0.76)	0.72 (0.58–0.84)	0.205
Reader 3 (specialist)	83.3 (61.2–94.6)	94.4 (74.8–99.6)	0.617	36.1 (22.4–52.3)	52.8 (37.0–68.0)	<0.001	0.71 (0.55–0.85)	0.82 (0.71–0.92)	0.022
Reader 4 (specialist)	94.4 (74.8–99.6)	100 (83.0–100.0)	1.000	38.9 (24.7–55.1)	50.0 (34.5–65.5)	0.261	0.76 (0.63–0.88)	0.85 (0.75–0.94)	0.070
Average of non-specialists	77.8 (0–100.0)	94.4 (94.4–94.4)	0.374	40.3 (0–100.0)	31.9 (0–100.0)	0.226	0.56 (0.41–0.71)	0.68 (0.54–0.82)	0.003
Average of specialists	88.9 (66.6–100.0)	97.2 (61.9–1.00.0)	0.183	37.5 (23.7–51.3)	51.4 (36.0–66.8)	0.028	0.74 (0.60–0.88)	0.84 (0.73–0.94)	0.018
Average of all readers	83.3 (65.6–100.0)	95.8 (90.7–100.0)	0.103	38.9 (22.8–55.0)	41.7 (17.1–66.2)	0.725	0.65 (0.48–0.81)	0.76 (0.62–0.90)	0.001

* Numbers are percentages, and 95% confidence intervals are provided in parentheses. ** Numbers in parentheses are the 95% confidence intervals of the AUC values. *p* values refer to the differences in diagnostic performance between the classic AB-MRI and modified AB-MRI protocols. AUC = area under the receiver operating characteristic curve; *p* = *p* value; AB-MRI = abbreviated breast magnetic resonance imaging; classic = classic AB-MRI; modified = modified AB-MRI. We classified the lesion size on MRI into two groups (>10 mm vs. ≤10 mm).

**Table 5 diagnostics-14-00282-t005:** Diagnostic performance of the four readers in assessing 122 breast lesions with the classic and modified abbreviated MRI protocols in relation to lesion type.

	Sensitivity *			Specificity *			AUC **		
	Classic	Modified	*p*	Classic	Modified	*p*	Classic	Modified	*p*
Focus									
Reader 1 (non-specialist)	66.7 (21.4–94.5)	66.7 (21.4–94.5)	NA	58.1 (40.8–73.6)	32.3 (18.4–49.7)	<0.001	0.66 (0.26–0.98)	0.44 (0.05–0.71)	0.001
Reader 2 (non-specialist)	66.7 (21.4–94.5)	66.7 (21.4–94.5)	NA	71.0 (53.6–84.1)	48.4 (32.0–65.1)	0.016	0.66 (0.16–0.98)	0.59 (0.05–1.00)	0.173
Reader 3 (specialist)	33.3 (5.5–78.6)	0 0.0–54.1)	0.999	87.1 (71.4–95.2)	90.3 (75.4–97.0)	1.000	0.56 (0.31–0.92)	0.45 (0.40, 0.50)	0.506
Reader 4 (specialist)	0 (0.0–54.1)	0 (–0.0–54.1)	NA	87.1 (71.4–95.2)	90.3 (75.4–97.0)	0.998	0.61 (0.36–0.84)	0.57 (0.37, 0.86)	0.748
Average of non-specialists	66.7 (1.3–100.0)	66.7 (1.3–100.0)	NA	64.5 (35.4–93.6)	40.3 (6.0–74.7)	0.002	0.67 (0.01–1.00)	0.67 (0.01–1.00)	0.319
Average of specialists	16.7 (0–100)	0	0.500	87.1 (77.0, 97.2)	90.3 (81.9, 98.7)	0.483	0.17 (0.00–1.00)	0	0.258
Average of all readers	41.7 (0–94.2)	33.3 (0–89.5)	0.391	75.8 (55.3–96.3)	65.3 (19.9–100.0)	0.300	0.42 (0.00–0.94)	0.33 (0.00–0.90)	0.071
Non-mass enhancement									
Reader 1 (non-specialist)	77.8 (45.9–94.3)	88.9 (57.3–98.9)	0.999	36.4 (22.1–53.3)	24.2 (12.6–40.8)	0.368	0.51 (0.32–0.69)	0.54 (0.37–0.71)	0.764
Reader 2 (non-specialist)	55.6 (26.8–81.2)	100.0 (71.2–101.1)	0.024	51.5 (35.2–67.5)	45.5 (29.8–62.0)	0.943	0.48 (0.29–0.69)	0.73 (0.59–0.86)	0.035
Reader 3 (specialist)	77.8 (45.9–94.3)	88.9 (57.3–98.9)	1.000	42.4 (27.2–59.1)	54.5 (38.0–70.2)	0.134	0.71 (0.49–0.92)	0.80 (0.62–0.95)	0.166
Reader 4 (specialist)	100.0 (71.2–101.1)	100.0 (71.2–101.1)	NA	45.5 (29.8–62.0)	54.5 (38.0–70.2)	0.568	0.83 (0.69–0.94)	0.85 (0.72–0.94)	0.742
Average of non-specialists	66.7 (0–100.0)	94.4 (23.9–100.0)	0.344	43.9 (7.0–80.9)	34.8 (0–100.0)	0.260	0.67 (0.00–1.00)	0.94 (0.24–1.00)	0.360
Average of specialists	88.9 (66.6–100.0)	94.4 (23.9–100.0)	0.500	43.9 (29.3–58.6)	54.5 (39.5–69.5)	0.044	0.89 (0.00–1.00)	0.94 (0.24–1.00)	0.352
Average of all readers	77.8 (51.3–100.0)	94.4 (83.0–100.0)	0.186	43.9 (31.5–56.4)	44.7 (23.8–65.6)	0.916	0.78 (0.51–1.00)	0.94 (0.83–1.00)	0.182
Mass									
Reader 1 (non-specialist)	87.5 (64.5–97.0)	100.0 (81.3–100.7)	0.617	26.7 (14.0–44.3)	16.7 (7.1–33.3)	0.424	0.62 (0.50–0.78)	0.75 (0.62–0.87)	0.131
Reader 2 (non-specialist)	81.2 (57.4–93.8)	87.5 (64.5–97.0)	1.000	46.7 (30.2–63.8)	43.3 (27.3–60.7)	1.000	0.71 (0.53–0.87)	0.69 (0.54–0.82)	0.836
Reader 3 (specialist)	87.5 (64.5–97.0)	100.0 (81.3–100.7)	0.617	50.0 (33.2–66.8)	63.3 (45.6–78.2)	0.368	0.75 (0.60–0.88)	0.86 (0.75–0.95)	0.072
Reader 4 (specialist)	75.0 (50.9–90.2)	87.5 (64.5–97.0)	0.617	63.3 (45.6–78.2)	70.0 (52.3–83.5)	0.868	0.76 (0.61–0.89)	0.88 (0.77–0.96)	0.032
Average of non-specialists	84.4 (44.7–100.0)	93.8 (14.3–100.0)	0.205	36.7 (0–100.0)	30.0 (0–100.0)	0.295	0.66 (0.50–0.87)	0.72 (0.60–0.84)	0.596
Average of specialists	81.2 (57.2–100.0)	93.8 (14.3–100.0)	0.025	56.7 (31.1–82.2)	66.7 (50.3–83.1)	0.186	0.76 (0.62–0.89)	0.87 (0.78–0.95)	0.014
Average of all readers	82.8 (69.4–96.2)	93.8 (83.3–100.0)	0.022	46.7 (24.6–68.7)	48.3 (12.0–84.7)	0.769	0.71 (0.58–0.84)	0.79 (0.66–0.92)	0.122

* Numbers are percentages, and the 95% confidence intervals are in parentheses. ** Numbers in parentheses are the 95% confidence intervals of the AUC values. *p* values refer to the differences in the diagnostic performance between classic and modified AB-MRI protocols. AUC = area under the receiver operating characteristic curve; *p* = *p* value; AB-MRI = abbreviated breast magnetic resonance imaging; classic = classic AB-MRI; modified = modified AB-MRI. We classified the lesion types on MRI into three groups (focus, non-mass enhancement, and mass).

**Table 6 diagnostics-14-00282-t006:** Diagnostic performance of four readers in assessing 122 breast lesions with the classic and modified abbreviated MRI protocols in relation to the degree of background parenchymal enhancement.

	Sensitivity *			Specificity *			AUC **		
	Classic	Modified	*p*	Classic	Modified	*p*	Classic	Modified	*p*
Minimal									
Reader 1 (non-specialist)	80.0 (49.6–94.9)	90.0 (60.4–99.0)	1.000	50.0 (34.5–65.5)	22.2 (11.5–37.9)	<0.001	0.66 (0.46–0.84)	0.62 (0.42–0.80)	0.750
Reader 2 (non-specialist)	80.0 (49.6–94.9)	80.0 (49.6–94.9)	0.617	44.4 (29.5–60.4)	41.7 (27.1–57.7)	1.000	0.61 (0.42–0.79)	0.64 (0.44–0.83)	0.800
Reader 3 (specialist)	70.0 (40.1–89.6)	70.0 (40.1–89.6)	0.617	55.6 (39.6–70.5)	66.7 (50.5–79.9)	0.261	0.68 (0.50–0.85)	0.72 (0.54–0.89)	0.489
Reader 4 (specialist)	60.0 (31.5–83.4)	60.0 (31.5–83.4)	NA	66.7 (50.5–79.9)	72.2 (56.2–84.3)	0.868	0.72 (0.54–0.88)	0.75 (0.57–0.91)	0.432
Average of non-specialists	80.0 (64.0–96.0)	85.0 (62.8–100.0)	0.500	47.2 (34.3–60.1)	31.9 (0–100.0)	0.417	0.64 (0.48–0.79)	0.63 (0.50–0.76)	0.869
Average of specialists	65.0 (35.2–94.8)	65.0 (35.2–94.8)	NA	61.1 (39.8–82.4)	69.4 (53.9–85.0)	0.148	0.70 (0.52–0.88)	0.74 (0.56–0.92)	0.088
Average of all readers	72.5 (50.2–94.8)	75.0 (52.1–97.9)	0.618	54.2 (39.4–68.9)	50.7 (16.0–85.4)	0.718	0.67 (0.51–0.82)	0.68 (0.53–0.84)	0.583
Mild									
Reader 1 (non-specialist)	88.9 (57.3–98.9)	88.9 (57.3–98.9)	0.617	29.2 (14.7–49.0)	20.8 (9.0–40.2)	0.803	0.55 (0.35–0.73)	0.72 (0.51–0.91)	0.118
Reader 2 (non-specialist)	55.6 (26.8–81.2)	100.0 (71.2–101.1)	0.024	62.5 (42.8–79.0)	50.0 (31.4–68.6)	0.716	0.60 (0.35–0.85)	0.76 (0.63–0.87)	0.284
Reader 3 (specialist)	66.7 (35.8–88.3)	88.9 (57.3–98.9)	0.617	58.3 (38.9–75.6)	70.8 (51.0–85.3)	0.568	0.66 (0.44–0.87)	0.80 (0.63–0.93)	0.106
Reader 4 (specialist)	88.9 (57.3–98.9)	100.0 (71.2–101.1)	1.000	50.0 (31.4–68.6)	62.5 (42.8–79.0)	0.424	0.76 (0.57–0.92)	0.85 (0.72–0.94)	0.254
Average of non-specialists	72.2 (0–100.0)	94.4 (23.9–100.0)	0.500	45.8 (0–100.0)	35.4 (0–100.0)	0.126	0.58 (0.42–0.73)	0.74 (0.66–0.82)	0.026
Average of specialists	77.8 (31.5–100.0)	94.4 (23.9–100.0)	0.172	54.2 (33.1–75.2)	66.7 (49.8–83.6)	0.070	0.71 (0.51–0.92)	0.82 (0.70–0.95)	0.114
Average of all readers	75 (48.1–100.0)	94.4 (83.0–100.0)	0.165	50.0 (28.1–71.9)	51.0 (19.0–83.1)	0.890	0.65 (0.46–0.83)	0.78 (0.67–0.89)	0.070
Moderate									
Reader 1 (non-specialist)	75.0 (31.0–96.3)	100.0 (52.8–101.8)	1.000	41.9 (26.4–59.2)	29.0 (15.9–46.4)	0.453	0.61 (0.32–0.87)	0.73 (0.54–0.90)	0.253
Reader 2 (non-specialist)	75.0 (31.0–96.3)	100.0 (52.8–101.8)	1.000	64.5 (47.1–79.0)	48.4 (32.0–65.1)	0.400	0.77 (0.51–0.96)	0.87 (0.71–0.98)	0.206
Reader 3 (specialist)	100.0 (52.8–101.8)	100.0 (52.8–101.8)	NA	64.5 (47.1–79.0)	67.7 (50.3–81.6)	1.000	0.91 (0.77–1.00)	0.92 (0.82–1.00)	0.780
Reader 4 (specialist)	100.0 (52.8–101.8)	100.0 (52.8–101.8)	NA	74.2 (56.9–86.5)	74.2 (56.9–86.5)	0.803	0.92 (0.81–0.99)	0.94 (0.84–1.00)	0.735
Average of non-specialists	75.0 (75.0–75.0)	100 (100.0–100.0)	0	53.2 (0–100.0)	38.7 (0–88.0)	0.062	0.69 (0.39–0.99)	0.80 (0.52–1.00)	0.010
Average of specialists	100 (100.0–100.0)	100 (100.0–100.0)	NA	69.4 (49.9–88.8)	71.0 (54.3–87.7)	0.743	0.92 (0.80–1.00)	0.93 (0.83–1.00)	0.791
Average of all readers	87.5 (64.5–100.0)	100 (NA–NA)	0.182	61.3 (40.9–81.7)	54.8 (24.7–84.9)	0.332	0.80 (0.59–1.00)	0.87 (0.70–1.00)	0.095
Marked									
Reader 1 (non-specialist)	80.0 (38.5–97.3)	100.0 (58.1–101.6)	1.000	0 (–2.0–54.1)	33.3 (5.5–78.6)	1.000	0.70 (0.30–1.00)	0.80 (0.60–1.00)	0.663
Reader 2 (non-specialist)	80.0 (38.5–97.3)	80.0 (38.5–97.3)	0.617	66.7 (21.4–94.5)	33.3 (5.5–78.6)	1.000	0.80 (0.47–1.00)	0.63 (0.30–0.90)	0.489
Reader 3 (specialist)	100.0 (58.1–101.6)	100.0 (58.1–101.6)	NA	66.7 (21.4–94.5)	100.0 (45.9–100.0)	1.000	0.93 (0.73–1.00)	1.00 (1.00–1.00)	0.340
Reader 4 (specialist)	60.0 (23.4–88.6)	80.0 (38.5–97.3)	1.000	66.7 (21.4–94.5)	100.0 (45.9–100.0)	1.000	0.77 (0.40–1.00)	0.97 (0.87–1.00)	0.224
Average of non-specialists	80.0 (80.0–80.0)	90.0 (0–100.0)	0.500	33.3 (0–100.0)	33.3 (0–98.7)	1.000	0.75 (0.12–1.00)	0.72 (0.38–1.00)	0.844
Average of specialists	80.0 (0–80.0)	90.0 (0–100.0)	0.500	66.7 (1.3–100.0)	100.0 (100.0–100.0)	0.568	0.85 (0.51–1.00)	0.98 (0.77–1.00)	0.373
Average of all readers	80.0 (53.9–100.0)	90.0 (69.5–100.0)	0.182	50.0 (0–100.0)	66.7 (10.5–100.0)	0.706	0.80 (0.59–1.00)	0.85 (0.60–1.00)	0.738

* Numbers are percentages and 95% confidence intervals are in parentheses. ** Numbers in parentheses are the 95% confidence intervals of the AUC values. *p* values refer to the differences in the diagnostic performance between classic and modified AB-MRI protocols. AUC = area under the receiver operating characteristic curve. *p* = *p* value. AB-MRI, abbreviated breast magnetic resonance imaging; classic = classic AB-MRI protocol; modified = modified AB-MRI protocol. We dichotomized the background parenchymal enhancement into minimal/mild and moderate/marked background parenchymal enhancement.

## Data Availability

The data presented in this study are not publicly available due to personal health information privacy policy and ethical restrictions but deidentification data are available from the corresponding author on reasonable request.

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
