# Peer review of "Comparison of Diagnostic Performance between Classic and Modified Abbreviated Breast MRI and the MRI Features Affecting Their Diagnostic Performance"

_diagnostics, 2024, doi:10.3390/diagnostics14030282_

Round 1

Reviewer 1 Report

Comments and Suggestions for Authors

This topic is really important since wider acceptance of abbreviated breast MRI would be an important step to increased access for women in both the screening and diagnostic settings. The modified version has many advantages over the classic version as you describe, both in length of examination and also increased specificity and specificity.     Improved performance of all readers   with the modified AB -MRI was an important finding in this study and would be especially significant if it raised the accuracy of non-experienced as well as experienced readers.  Also, ACR accreditation for modified AB- MRI would be beneficial for wider acceptance and also   insurance approval in the U.S.

Author Response

Response: Since abbreviated breast MRI (AB-MRI) emerged in 2014, showing diagnostic accuracy and cancer-detection rates comparable to those of conventional full-protocol MRI, AB-MRI protocols have not been standardized. Classic AB-MRI does not fulfill the ACR breast MRI accreditation requirements despite several advantages. To fulfill the ACR breast MRI accreditation requirements and overcome the several limitations of the classic AB-MRI protocol, we implemented a modified AB-MRI protocol. Therefore, the purpose of this study was to compare the diagnostic performance of the modified AB-MRI protocol with that of the classic AB-MRI and to determine the MRI features affecting the diagnostic performance of both. Improved performance of all readers with the modified AB-MRI was an important finding in this study and would be especially significant if it raised the accuracy of non-experienced as well as experienced readers. Although further studies are needed, considering the results of the diagnostic performance of classic AB-MRI and modified AB-MRI and the MRI feature affecting the diagnostic performance of AB-MRI in this study could enable the application of optimized and individualized AB-MRI techniques, leading to improved diagnostic performance in the near future.

Reviewer 2 Report

Comments and Suggestions for Authors

1. Was the single-center retrospective design subject to bias compared to a prospective multi-center approach? How were case selection and analysis standardized? 

2. What was done to account for and analyze inter-reader variability, given differences in interpretation could impact reported accuracy results?  

3. How were the classic and modified MRI protocols anonymized and presented to readers? Could subtle differences have still introduced bias into assessments? 

4. Why were only lesions scoring BI-RADS 4-5 on MRI and ultrasound included? Could excluding category 3 findings affect sensitivity/specificity calculations? 

5. What was the follow-up period for lesions deemed benign based on biopsy or surveillance? Could longer term outcomes like interval cancers differ between protocols? 

6. How were variability in individual patient factors (e.g. breast density), equipment, and study protocols between sites controlled for in the analysis? 

7. On what factors was histopathology prioritized as the reference standard over longer term clinical or imaging follow-up for some lesions?

8. What was the rationale for using only two kinetic curve categories in the analysis? Could a more granular approach provide additional diagnostic value?

9. Without a cost-effectiveness analysis, how can the added value of the modified protocol be weighed against longer exam times on a population scale?

Author Response

We thank the reviewers for their thoughtful suggestions and insights, which have enriched

the manuscript and produced a better and more balanced account of the research.

  1. Was the single-center retrospective design subject to bias compared to a prospective multi-center approach? How were case selection and analysis standardized?

Response: The single-center retrospective design is a limitation of this study. However, we have made an effort to minimize bias by analyzing a relatively large number of breast lesions. Regarding case selection and analysis, all lesions categorized as BI-RADS 4 or 5 on MRI scans performed between August 2019 and January 2022 were included and all readers, who were blinded to the patients’ clinical information, images from other modalities, and histopathological results of the breast lesions, individually reviewed the anonymized image files. A future multicenter study would be helpful in further validating our results.

  1. What was done to account for and analyze inter-reader variability, given differences in interpretation could impact reported accuracy results? 

Response: In response to concerns about how differences in interpretation could affect reported accuracy results, we have added the interobserver agreement for classic and modified AB-MRI in the Materials and Methods and Results sections.

“The readers agreement on BI-RADS final assessment based on both classic and modified AB-MRI was assessed using kappa (κ) statistics, and we used the following definitions: less than 0.20 indicates poor agreement; 0.21–0.40 indicates fair agreement; 0.41–0.60 indicates moderate agreement; 0.61–0.80 indicates good agreement; and 0.81–1.00 indicates very good agreement [14].” (page 5, lines 175–179)

“The overall interobserver agreement for the BI-RADS final assessment based on modified AB-MRI (k=0.574, 95% confidence interval [CI]:0.551-0.586) was higher than those based on classic AB-MRI (k=0.538, 95% CI:0.524-0.551).” (page 5, lines 199-201)

  1. How were the classic and modified MRI protocols anonymized and presented to readers? Could subtle differences have still introduced bias into assessments?

Response: In the '2.3. Breast MRI Analysis' section on page 4 of our manuscript, we detailed how we employed the anonymization feature within the PACS system to extract MRI images. These images were subsequently stored in a designated research folder and presented to the readers. To reduce potential bias, we reorganized the anonymized, modified breast MRI scan images in a random order and presented them to the readers six weeks (washout period) after providing them with the classic breast MRI scan images.

  1. Why were only lesions scoring BI-RADS 4-5 on MRI and ultrasound included? Could excluding category 3 findings affect sensitivity/specificity calculations?

Response: The purpose of this study was to evaluate the diagnostic performance in distinguishing between benign and malignant lesions detected on MRI. Given that the readers were already informed about the patients' history of breast cancer surgery, it was considered that including BI-RADS category 3 lesions could lead to an increased rate of false positives. This inclusion was anticipated to pose challenges in accurately measuring the diagnostic performance. Therefore, to ensure a more precise evaluation, category 3 lesions were excluded from our analysis. There is a study with a similar design in support of our methodological approach [1], and it is also noted in the manuscript as reference 14 that the findings of this study are consistent with our results. Another previous study reported that including BI-RADS category 3 lesions on breast MRI may have a minor impact on the results due to the low malignancy likelihood and approximate 100% NPV of BI-RADS category 3 lesions on breast MRI [2]. Therefore, although there may be an increase in false positive rates, the difference of diagnostic performance between classic and modified AB-MRI may have been minimally impacted. We added the content in the limitation section as follows.

“Fourth, breast lesions assigned as BI-RADS category 4 or 5 on breast MRI or US were included in this study. Although these lesions might lead to overestimating sensitivity of breast MRI, the difference in diagnostic performance between classic and modified AB-MRI is likely to have been minimally impacted.” (page 14, lines 312-315)

References

  1. Kim, E.S.; Cho, N.; Kim, S.Y.; et al. Comparison of abbreviated MRI and full diagnostic MRI in distinguishing between benign and malignant lesions detected by breast MRI: a multireader study. Korean J Radiol 2021, 22, 297–307.
  2. Xie Z; Xu W; Zhang H.; et al. The value of MRI for downgrading of breast suspicious lesions

 detected on ultrasound. BMC Med Imaging 2023,23,72.

  1. What was the follow-up period for lesions deemed benign based on biopsy or surveillance? Could longer term outcomes like interval cancers differ between protocols?

Response: After receiving benign results based on histopathologic confirmation, we established semi-annual surveillance intervals for patients, alternating breast US and AB-MRI, for at least one and half years because we collected their medial chart review and imaging data in August 2023. Furthermore, we reviewed the final outcomes of patients with benign lesions. Although long-term follow-up is necessary to accurately evaluate the diagnostic performance of AB-MRI, no interval cancers have been detected in either classic or modified AB-MRI to date. We added the content in the limitation section as follows.

“Fifth, patients who received benign results based on histopathologic confirmation underwent relatively short-term follow-up imaging surveillance. To accurately evaluate the diagnostic performance of AB-MRI, further studies involving long-term follow-up imaging surveillance will be necessary.” (page 14, lines 315-318)

  1. How were variability in individual patient factors (e.g. breast density), equipment, and study protocols between sites controlled for in the analysis?

Response: All lesions in women with a personal history of breast cancer surgery categorized as BI-RADS 4 or 5 on MRI scans performed between August 2019 and January 2022 were included without controlling for variability in the analysis. In our study population of 122 cases, the distribution of breast density was: category A, 1.6% (2 cases); category B, 15.6% (19 cases); category C, 62.3% (76 cases); and category D, 20.5% (25 cases). Equipment and study protocols were consistent across all cases, thus precluding the need for further controls. For accurate analysis, we plan to control for variability in future studies as your suggestion.

  1. On what factors was histopathology prioritized as the reference standard over longer term clinical or imaging follow-up for some lesions?

Response: While histopathology is a more accurate gold standard reference compared to long-term clinical or imaging follow-up, previous studies have categorized breast lesions without histological confirmation as benign if they showed negative findings in one-year follow-up images [1, 2]. Consequently, in our study, lesions confirmed by histopathology were classified as malignant or benign accordingly. However, for lesions without histopathological confirmation but stable in one-year follow-up, we classified them as benign based on the follow-up results. Additionally, as mentioned in response to the fourth comment, only lesions categorized as BI-RADS 4 or 5 were included in our study population to reduce the false positive rate.

References

  1. Park, K.W.; Han, S.B.; Han, B.K.; et al. MRI surveillance for women with a personal history of breast cancer: comparison between abbreviated and full diagnostic protocol. Br J Radiol. 2020, 93, 20190733.
  2. Kwon, M.R.; Ko, E.Y.; Han, B.K.; et al. Diagnostic performance of abbreviated breast MRI for screening of women with previously treated breast cancer. Medicine (Baltimore). 2020, 99, e19676.

  1. What was the rationale for using only two kinetic curve categories in the analysis? Could a more granular approach provide additional diagnostic value?

Response: We appreciate your comment and believe these changes provide a more granular and insightful analysis. As per your suggestion, we revised the delayed phase kinetic curve categorization into three categories in the '2.3. Breast MRI Analysis' section and updated Table 2, as shown below. In our analysis of 122 cases: in the benign lesion group (94 cases), the findings were persistent, 27.7% (26 cases); plateau, 22.3% (21 cases); and washout, 50.0% (47 cases). In the malignant lesion group (28 cases), the results were persistent, 17.9% (5 cases); plateau, 10.7% (3 cases); and washout, 71.4% (20 cases). The Fisher's exact test showed a P value of 0.129. Since the readers reviewed the images on an INFINITT PACS monitor, the remained results were not affected. 

“Kinetic-curve findings of the delayed phase were divided into three groups for analysis: persistent, plateau, or washout, as defined by the ACR BI-RADS [10].” (page 5, lines 170-171)

Table 2. Comparison of AB-MRI characteristics between benign and malignant breast lesions (page 6)

  1. Without a cost-effectiveness analysis, how can the added value of the modified protocol be weighed against longer exam times on a population scale?

Response: We did not perform a cost-effectiveness analysis, and therefore cannot assess the exact effectiveness of the modified abbreviated breast MRI compared to the classic breast MRI. However, with its clear time-saving advantage, the modified protocol is expected to enhance patient comfort during MRI exams and to yield diagnostic performance comparable to the full protocol. Thus, further study on the cost-effectiveness of modified protocol would be beneficial for more comprehensively validating our results. We have addressed this aspect in the limitations section of our manuscript in response to your comments as follows.

“Finally, a cost-effectiveness analysis of the modified AB-MRI protocol was not performed, suggesting a need for further study to fully assess its effectiveness compared to the classic protocol.” (page 14, lines 385–388)

We look forward to hearing from you and would be happy to make further changes, if required.

Round 2

Reviewer 2 Report

Comments and Suggestions for Authors

very well. I hope my comments and suggestions have been useful for you.